# A Comparative Analysis of Orexins in the Physio-Pathological Processes of the Male Genital Tract: New Challenges? A Review

**DOI:** 10.3390/vetsci11030131

**Published:** 2024-03-15

**Authors:** Anna Costagliola, Luigi Montano, Emilia Langella, Renato Lombardi, Caterina Squillacioti, Nicola Mirabella, Giovanna Liguori

**Affiliations:** 1Department of Veterinary Medicine and Animal Productions, University of Napoli Federico II, 80137 Naples, Italy; caterina.squillacioti@unina.it (C.S.); nicola.mirabella@unina.it (N.M.); giovanna.liguori@unina.it (G.L.); 2Andrology Unit of the “San Francesco d’Assisi” Hospital—ASL Salerno, EcoFoodFertility Project Coordination Unit, Oliveto Citra, 84020 Salerno, Italy; luigimontano@gmail.com; 3PhD Program in Evolutionary Biology and Ecology, University of Rome “Tor Vergata”, 00133 Roma, Italy; 4School of Agricultural, Forestry, Food and Environmental Sciences (SAFE), University of Basilicata, 85100 Potenza, Italy; emilia.langella@unibas.it; 5Local Health Authority, ASL Foggia, 71121 Foggia, Italy; renato.lombardi@aslfg.it

**Keywords:** orexins, hypocretin, orexin receptors, male genital tract, rodents, ruminants, dog

## Abstract

**Simple Summary:**

Orexins A (OXA) and B (OXB) are two peptides first identified in the hypothalamus that act by binding two specific receptors, receptor 1 (OX1R) and 2 (OX2R), for orexins. They are widely distributed in different organs and behave as pleiotropic regulators of a wide diversity of biological functions, including sleep, excitement, nutrition, reward, circadian rhythm, anxiety, cognition, and reproduction. Here we examine the experimental evidence collected in recent years to support the male reproductive aspects of orexins, with particular emphasis on our current knowledge of their expression patterns and potential functional roles in physiological and pathological conditions. Overall, the available data strongly suggest that orexins can operate as neuroendocrine regulators on spermatogenesis and steroidogenesis. We discuss the richness of the functions associated with the orexinergic system and hypothesize the potential of such peptides as a therapeutic goal for a number of reproductive disorders and therefore the processing of drugs for the treatment of male hypo/infertility and cancer.

**Abstract:**

Orexins A (OXA) and B (OXB) and their specific receptors, receptor 1 (OX1R) and 2 (OX2R) for orexins, are hypothalamic peptides involved in orchestrating several functions in the central nervous system and peripheral organs, including sleep, excitement, nutrition, reward, circadian rhythm, anxiety, cognition, and reproduction. The aim of this narrative review is, in particular, to speculate the role of orexins in the male genital tract of animal species and human beings. The experimental evidence collected in recent years assumed that in the testes of the animal species here described, orexins are directly involved in steroidogenesis and spermatogenesis regulation. In the epididymis, these peptides are locally synthesized, thus suggesting their role governing the fertilizing capability of the immature male gamete. In addition to playing a physiological role, orexins are involved in numerous inflammatory and/or neoplastic pathologies too. The expression of the orexinergic system in prostate cancer suggests that they might play a potential therapeutic function. Overall, the future directions of this literature review allow us to hypothesize a role of the orexinergic complex not only as a marker for the diagnosis of certain tumors affecting the male genital tract but also for the treatment of hypo/infertility condition.

## 1. Introduction

Orexins (OXs) and their receptors and HCRT (hypocretins)-producing cell bodies constitute the OX/HACRT system. OXs bind two G protein-coupled receptors—orexin receptor 1 (OX1R) and orexin receptor 2 (OX2R)—with a 64% homology in their amino acid level [1]. The two receptors are highly conserved between mammalian species, showing 94% and 95% homology in rats and humans [1]. The system has been described in the central nervous system [2,3], as well as in peripheral organs such as the pituitary gland, kidney, and adrenal gland [4]. OXs mediate several functions in mammals, including humans. OXs comprise orexin A (OXA) and orexin B (OXB), both of which derive from a 130-amino-acid precursor peptide, prepro-orexin (PPO), by proteolytic cleavage. Such binding activates the PLC/IP_3_ pathway. OX1R binds OXA; in contrast, OX2R binds both OXA and OXB with similar affinity. The system can regulate several systems and functions, such as the sleeping–wakefulness cycle, emotion, food intake, behavior, energy metabolism, immunity, food intake, glucose metabolism, energy expenditure, and obesity [2,3]. OXs also participate in reproduction (both in male and female animal species) through an integrated neuroendocrine process at the central and peripheral level [5]. In particular, OXA regulates the activity of gonadotropin-releasing hormone (GnRH) neurons and gonadotropin-secreting pituitary which, in turn, interact with the hypothalamic–pituitary–gonadal (HPG) axis cells, as well as acting directly at the gonadal level, contributing to the control of sexual maturity and fertility [6].

The OX/HCRT system has mainly been described in laboratory animals [7,8,9,10,11,12,13,14,15,16] and in humans, both in healthy and cancerous stages, using different techniques [17,18,19,20].

The present review aims to summarize state-of-the-art research, providing a comparative analysis on the presence of and role that the orexinergic system plays in the male genital tract of animals, including humans. Therefore, articles published between 1998 and 2023 were considered by the authors independently, using five strings (related to our topic) on PubMed, Web of Science, SCOPUS, and Google Scholar.

Next, this narrative review also identifies gaps in the literature and suggests new research insights regarding the effects of the orexinergic system in the treatment of some types of neoplasia caused by environmental contaminants. A very innovative aspect is discussed which, once addressed, may facilitate the clinical translatability of neuroendocrine–orexin interactions from experimental animals to human beings. This narrative review intends to summarize (1) pharmacotherapy targeting the orexinergic system in clinical andrology and (2) related studies which are increasingly aware of the environmental influences on the onset of male infertility and cancer.

The purpose of this review article is to identify gaps within the literature in order to promote other and more specific therapeutic applications.

## 2. The Orexinergic System in the Human Male Genital Tract

PPO expression has been observed in the human testis, epididymis, prostate [19], and penis [17]. The schematic representation of the topographic distribution of the orexinergic complex is summarized in Table 1 and Table 2.

OX1R and OX2R have been localized in Leydig cells (which produce testosterone), in certain Sertoli cells, and in testicular peritubular myoid cells (TPMCs), as well as in the epididymis, seminal vesicles, and penis [17]. The expression of orexin receptors in Leydig cells is consistent with the role of testosterone (T) as found in the rat adrenal system at the mRNA level [21]. The effects of OXs on steroidogenesis are also known. In humans, adreno-cortex OXs stimulate corticosterone and cortisol production through the activation of the adenylate cyclase-dependent signaling cascade [22]. Therefore, orexins, by linking their specific receptors on Leydig cells, might regulate T production that, in turn, can regulate receptor expression in tissues, creating a positive feedback loop. Therefore, further investigations are necessary to determine the precise role/s of OXs on Leydig cells. Orexin receptors are localized on TPMCs too. TPMCs are contractile cells that express the cytoskeletal markers of smooth muscle cells [23]. Seminiferous tubule contraction regulates spermatogenesis and testicular sperm output. Karteris et al. [17] demonstrated that OXs were able to induce IP3 production from testicular membrane preparations by the activation of the PLC pathway, with OXB being more potent. Thus, orexin receptors can couple to Gq family members, activating the PLC/PKC/IP3 system. Angiotensin II can contract rat TPMCs via a protein kinase C-dependent mechanism [23]. Other hormones, such as endothelin-1, angiotensin II, and oxytocin, also act through G protein-coupled receptors, modulating the TPMC function in an endocrine/paracrine/autocrine manner [23,24,25]. Due to their signaling similarities, it might be supposed that OXs, too, can modulate the contractile state of TPMCs through the activation of PLC/IP3, providing structural integrity to seminiferous tubules. In contrast, no expression of OXA was observed in human samples [17,26]. The scattered expression of orexin receptors has been observed in Sertoli cells, suggesting their possible implication in signals regulating the rate of new sperm cell production.

Future studies should show how OXs are involved in spermatogenesis. The presence of orexin receptors in the human penis and seminal vesicles indicates a putative role in the control of penile function. Nitric oxide (NO) contributes to maintaining penile erection [27]. NO is produced by the catalytic oxidation of L-arginine by the enzyme NO synthase (NOS). NOS occurs in three distinct isoforms: two constitutive Ca2-dependent isoforms—endothelial NOS (eNOS) and neuronal NOS (nNOS)—and the Ca2-independent inducible NOS [28]. The Ca2-dependent nNOS has been found in the human penis [29]. In the small intestine of mice, the NO synthesis inhibitor *N*Gnitro- l-arginine completely inhibited the muscle relaxation induced by exogenous OXA [30]. Thus, OXs expressed in the penis might influence erectile function, modulating NO production by a Ca2-dependent NOS isoform. Orexin receptors are expressed in human Leydig cells, suggesting their implication in steroidogenesis. OXs are also expressed in TPMCs, where they might activate the PLC/IP3 cascade, promoting further testicular function. Last, orexin receptors expressed in Sertoli cells, epididymis, and seminal vesicles further suggest differential effects in each tissue by the activation of multiple signaling pathways. Future, more extensive investigations should look at the potential roles of OXs on steroidogenesis and on the whole control of the reproductive system.

Contrasting results have been reported in the human prostate. Nakabayashi et al. (2003) [26] failed to detect PPO mRNA expression and OXA localization in the prostate of healthy subjects, while Valiante et al. [19] detected the presence of OXA and OX1R in normal and hyperplastic prostates. In particular, OXA and OX1R were detected in a sub-population of the exocrine epithelium but not endocrine (the latter being Chromogranin immunopositive) in all normal prostates and, more abundantly, in hyperplastic prostates. PPO and OX1R were also expressed in a human prostate cell line PNT1A and in human prostate cancer (PCa) from lower to higher grades of malignancy [20]. In contrast, Malendowicz et al. [31] demonstrated the presence of OXA and OX2R, but not PPO and OX1R expression, in normal and hyperplastic prostate.

Alexandre et al. [18] reported the presence of OXA and OX1R in two cancer cell lines, the androgen-responsive cell line LNCaP and the androgen-unresponsive cell line AI DU 145, in benign prostate hyperplasia (BPH) and in cancer of the prostate (CaP) at various stages. In contrast, OXA was not detected in the CaP foci. In the advanced CaP, the adenocarcinomatous formations largely expressed OX1R. OX2R was detected only in a few cancer cells in advanced CaP and OX1R in scattered cells of the BPH tissues. The OX1R gene was found in the androgen-unresponsive cell line DU 145, but not in the androgen-responsive cell line LNCaP, even in the presence of db-cAMP (1 mM)/3-isobutyl-1-methylxanthine (IBMX), a substance that induces neuroendocrine differentiation and mediates apoptosis [18] (normal and pathological localizations are reported in Table 1 and Table 2).

The different methodological procedures or the high turnover of internalization and/or cellular production of OXs and their receptors might have caused a discrepancy between the results of Valiante et al., Alexandre et al., and Costagliola et al. [18,19,20,32].

## 3. The Orexinergic System in the Male Genital Tract of Rodents

### 3.1. Rat Male Genital Tract

*Testis:* This organ expresses PPO mRNA [12,16,21,33]. More specifically, OXA peptide has been shown to be localized in Leydig cells and spermatocytes during the development of the germinative epithelium, with the exception of the V and VI stages [16] in Sertoli cells, and in rounded and oblong spermatids, in the second half of the cycle [12]. OX1R [34], but not OX2R [12,16,35], has also been observed in developing spermatocytes and spermatids of the seminiferous tubules of rat testes, where the highest levels of PPO and OX1R mRNAs have been found [16]. In rat testes, the expression of OXs is age-dependent, with the max expression occurring in adulthood [35]. OXA may play a paracrine role too, since OX1R is expressed on germinal cytotypes both in humans [17] and rats [35]. Thus, OXA, produced by Sertoli cells, could act on germ cells, but their role is still unknown. Nevertheless, the endocrine diffusion of OXA in the testis cannot be excluded, especially in the interstice, which is rich in blood vessels. These data confirm the multiple regulatory activities played by OXs via (i) the hypophyseal gonadotropins that regulate steroidogenesis and sperm cell development and (ii) a pool of substances (including OXA) produced by the testis and that regulate testicular functions acting locally [6].

Treatment of rat testicular slides with LH induced a significant increase in the T treatment of rat testicular slides, with OXA alone also increasing T secretion. When Müllerian inhibiting substance (MIS) was added to OXA-treated slices, the basal hormone production was almost restored. The addition of MIS alone decreased the production (by about 70%) of T. When OXA was added to MIS-treated slices, the inhibitory activity of MIS on T secretion was lowered by about 30%. Under these experimental conditions, LH still stimulated T production. The selective OX1R inhibitor SB-408124 inhibited OXA, stimulating the production of T. When SB-408124 was added alone, this had no effect on T production. Thus, OXA, when binding to OX1R, is able to counteract the inhibitory activity of MIS on T production. These experiments confirm the existence of crosstalk between OXA and MIS in the regulation of T synthesis in the male gonad. In addition, OXA may regulate steroidogenic activity when produced by Sertoli cells, acting on the same cells in an autocrine way, as well as may inhibit the production of the stem cell factor (SCF) which, in turn, stimulates the proliferation of the germinative epithelium [16]. Thus, the seminiferous tubules of mammalian testis contain different cell types which are capable of secreting and/or internalizing OXA; therefore, both autocrine and paracrine modalities of action can be supposed [6,34].

According to Barreiro et al. [35], E2 administration in rats during the neonatal period suppresses the OX1R mRNA level in the testes in adulthood.

Discrepancies observed between the results of previous studies [21,35,36] can be attributed to the high turnover of cellular production and/or internalization of OXs and their receptors. Opposing results on the localization of Oxs and their receptors have been reported in other peripheral organs too, such as the adrenal glands [37].

Furthermore, the use of different methodological experimental conditions cannot be excluded to test the ability of OXB to regulate testicular androgenesis in vitro [13]. In particular, the authors tested the testis T production after separate treatments with OXA, OXB, and LH (the latter was used to evaluate the responsiveness of the tissue used). In contrast to OXA, OXB did not affect T synthesis. In other organs, such the adrenal cortices of rats and humans, OXA enhanced glucocorticoid secretion, while OXB was either less potent or ineffective [38]. Although further studies are needed to fully establish the activity of OXB in the mammalian testis, these results demonstrate that while OXA induces steroidogenesis in rat testes [16,34,36], OXB is not directly involved in testicular steroidogenesis.

In addition to OXA [12], nitric oxide [39], aspartic acid [40], grelin [16], and leptin [41] are too produced by both the hypothalamic/hypophyseal system and testis. These substances may regulate, directly or indirectly, the production of gonadotropins. On this basis, it is likely that male gonad activity is under an autoregulatory mechanism in addition to gonadotropin control. Sertoli cells of the rat testis also produce OXA, strongly suggesting the involvement of the peptide in the control of seminiferous tubules [11]. Nevertheless, more extensive studies are necessary in order to establish the molecular mechanisms by which OXA act in the mammalian male genital tract.

*Epididymis:* Systemic OXA might arrive to the segment by the blood. OXA, by the cleavage of PPO, and OX1R have also been found in the vast majority of the principal cytotype of the epithelial cells lining the tubules in the caudal epididymis [11,17].

PPO and OX1R found in the rat epididymis are structurally similar to the peptides found in the brain or other mammalian organs [16,21,33,42,43,44]. The principal cells bordering the epididymal tubules are actively engaged in carrying electrolytes and water. These processes lead to the composition of a suitable fluid environment, in which spermatozoa obtain their fertilizing ability and motility [45]. Secretin and pituitary adenylate cyclase-activating peptide (PACAP) [46] *perform major roles* in controlling electrolyte transport in the epididymis. Similarly to secretin and PACAP, the peptide OXA presents apical localization in the principal cells of the epididymis and binds G-coupled receptors. The literature shows a combined action of OXA with the other two neurotransmitters (i.e., secretin and PACAP) in the regulation of the transepithelial electrolyte environment and fluid transport within the male genital tract. Thus, OXA may regulate the environment of epididymis, favoring the fertilizing ability of the immature spermatozoa (normal localizations are reported in Table 1).

The principal cells of the rat epididymis also revealed both OXB and OX2R immunoreactivity [47].

### 3.2. Murine Male Genital Tract

*Testis*: OXA peptide has been found to be localized in Sertoli cells, spermatogonia, spermatocytes, spermatids, and Leydig cells, while OX1R has been found to be localized in Leydig cells, spermatids, Sertoli cells, and spermatocytes, all with different intensities in the specific cell sub-types and along all stages of post-natal development, reaching the highest expression at 90 days post-partum [8,10]. OXB and OX2R have been described in the interstitial and tubular compartments of the testis, with a significant increase from post-natal to adult age [7]. In adult P mice, OXB and OX2R have been shown to be localized in the seminiferous tubules, in spermatocytes, spermatids, and Leydig cells in particular. OXB appeared transiently in Sertoli cells at 42 dpp, while OX2R never localized in Sertoli cells [7]. Thus, the OXB/OX2R bond in the testicle plays a determining role in regulating the proliferation and development of germ cells and Leydig cells until puberty, as well as being involved in controlling spermatogenesis, the functional maturation of Sertoli cells, and steroidogenesis within the adult murine testis [7].

Functional studies by Joshi et al. [9] illustrated the implication of the OXA/OX1R signaling circuit in regulating the glucose balance in the testis of adult mice. Treatment with an OX1R antagonist in adult mouse testes decreased the testicular glucose level, determining contemporary down-regulation in the expression levels of both GLUT3 and GLUT8. GLUT expression levels are associated with the tissue glucose consumption rate [48]. Hence, the reduced glucose uptake by testicular cells might be ascribed to reduced expression levels of GLUT3 and GLUT8 in the male gonad. GLUT8 is present intracellularly and undergoes translocation to the juxtamembrane when activated by an increase in glucose absorption [49]. The intracellular distribution of GLUT8 is comparable to that of GLUT4, an insulin-responsive glucose transporter [49,50]. Sertoli cells create an appropriate luminal milieu in the seminiferous tubules for the germ developing cycle [51] through the metabolization of glucose and its subsequent conversion to lactate [52]. To provide suitable lactate production, Sertoli cells determine a metabolic event, defined as the Warburg phenotype, which is representative of neoplastic cells, where much of the pyruvate produced in the course of glycolysis is transformed in lactate by lactate dehydrogenase (LDH) instead of undergoing oxidation through the tricarboxylic acid (TCA) cycle [53]. A substantial decrease in LDH enzyme activity in the male gonad was also induced in OX1R antagonist-treated mice. The decrease in the level of glucose and LDH activity in the male gonad after treating with OX1R antagonist inhibited lactate secretion, thus resulting in a sharp increase in testicular apoptosis, influencing germ cell proliferation and survival. Lactate is a glucose metabolite, which has been defined as the main source of germ cell sustenance [54]. It also has an anti-apoptotic reaction on the germline compartment [55]. In male gonads, lactate is not commonly defined as a compound of glycolysis but rather as a dynamic *substance involved in metabolism* and synthesized in significant quantities from Sertoli cells, serving as a fuel for germ cell development [56]. Thus, any interference with the glycemic metabolism of Sertoli cells may lead to a reduction in the production of lactate by germ cells, causing apoptosis of the latter [57]. Therefore, it has been hypothesized that a reduction in the concentration of glucose with the decreased activity of LDH compromises the synthesis of lactate in the testes treated with OX1R antagonist in mice, causing a clear pro-apoptotic action. As reported by Joshi [8,9,10], the interference of OXA–OX1R binding in the male gonads of adult mice is responsible for the reduction in the expression levels of GLUT3 and GLUT8, causing a decrease in glucose absorption along with reduced LDH activity in the testis, which could consequently jeopardize lactate production and affect the kinetics of germ cells. Joshi et al. [9] observed a marked increase in testicular germ cells in mice handled with antagonists rather than the controls, suggesting an increase in testicular apoptosis. Joshi et al. [9] also carefully evaluated the expression of numerous viz key proteins, including p53, Bax, Bcl-2, and caspase-3, which appear to play pivotal roles in the determinism of apoptosis [58]. The authors observed a marked increase in p53 and Bax protein expressions and a decrease in Bcl-2 peptide in the male gonads of mice treated with antagonists rather than the controls. Furthermore, an increment in the Bax/Bcl-2 ratio seems to be involved in the determination of apoptotic phenomena. The up-regulated expression of caspase-3 seems to demonstrate the responsibility of this peptide in the proteolytic action of various peptides, causing numerous morphological and biochemical alterations in apoptotic cells [59]. In addition, the expression of a cell proliferation marker known as proliferating cell nuclear antigen (PCNA) exhibited a significant decrease, as well as the number of PCNA-positive germ cells, in the testes of mice handled with antagonists, indicating a drop in the proliferative activity of germ cells. PCNA plays a pivotal role in DNA synthesis, cell growth, and proliferation, mostly during the late G1 and S phases [60]. PCNA was found to be more copious during the S phase and confirmed to regulate cell proliferation in germinative epithelium under the influence of several endocrine parameters [61]. The non-existence or lowest levels of PCNA in cells is a sign of apoptosis [57].

A low expression level of OX1R in adult mouse testes was observed during alloxan-induced T1D, which appeared to cause a reduction in the level of WT1 [8,10], a nuclear transcription factor that is predominantly expressed in Sertoli cells, which assumed a determining role in spermatogenesis by maintaining the polarity of Sertoli cells [62]. Thus, the OXA/OX1R system assumes a role in the down-regulation of WT1 expression, although the precise mechanism remains to be determined by Joshi et al. [9]. The OXA/OX1R system down-regulated the expression of WT1 in adult Sertoli cells, de-regulating numerous genes representing relevant signaling compounds in apical ectoplasmic specialization (ES)—a junction network between Sertoli cells and elongated spermatids—thus increasing apoptotic phenomena in germ cells [63].

Numerous peptides, such as OXA, neuropeptide Y, kisspeptin, GnRH, and GnIH/RFRPs, act at the testicular level [9,64,65,66], the functional roles of which remain to be fully determined, especially regarding glucose metabolism and regulation. Joshi et al. [9] suggested that these peptides appear to be involved in glucose metabolism in the male gonads of adult mice, supporting spermatogenesis in this regard. However, further studies are needed to deepen our understanding of the primary “apparatus” behind the OXA/OX1R complex-regulated modulation of testicular glucose metabolism.

As is known from the literature, OXA acts by binding two G protein-coupled receptors: OX1R and OX2R. In the presence of the antagonist SB-334867, which is specific to OX1R, there was a slightest chance for OXA to bind OX1R in antagonist-treated mice [67]. However, possible OXA/OX2R binding cannot be excluded, the role of which would require further investigation. A study by Joshi and Singh [67] showed that healing with an OX1R antagonist resulted in a sharp decrease in testicular weight associated with considerable degenerative processes of seminiferous tubules. A substantial drop in diameter, as well as marked increases in the percentage of affected seminiferous tubules and the tallness of the germinal epithelial layer in the testes of the antagonist-treated mice, were also observed, suggesting an adverse impact of the OX1R antagonist on spermatogenic process [67].

T is the hormone which is directly part of the regulation process of male fertility and, therefore, spermatogenesis [68]. The production of this substance at the testicular level requires the expression of multiple peptides, such as SF1, star, P450scc, 3b- and 17b-HSD, and 17a-hydroxylase [69,70]. As reported by Joshi and Singh [67], the reduction in T level following treatment with the OX1R antagonist could be ascribed to the down-regulation of the expression levels of SF1, star, P450scc, and 17b-HSD. Thus, treatment with the OX1R antagonist inhibited T production. According to Barreiro et al. [35], OXA may have a regulatory role on T biosynthesis in rat testes. Therefore, inhibiting the OXA/OX1R bond may lead to the down-regulation of the expression of a transcription factor called SF1, fundamental for the expression of all steroidogenic genes P450 [71]. SF1 intervenes in the expression of steroidogenic enzymes at the level of the adrenal cortex and gonads, thus modulating the release of cholesterol, which is a precursor for numerous steroidogenic reactions and modulates the expression of the StAR protein [69]. Therefore, it is possible that reduced SF1 expression levels after treatment with the OX1R antagonist may be responsible for the down-regulated expression levels of StARs and P450scc in mice. Contrarily, a sharp increase in the expression level of 3b-HSD in mice treated with antagonists has been observed, despite the down-regulated expression of SF1, which performs as a transcription factor for the expression of this enzyme [72]. Besides SF1, many other substances, such as Dax-1, fetoprotein transcription factor, GATA4 and 6, and LH/hcg, seem to affect and modulate the transcription and the efficiency of 3β-HSD [73]. In contrast, Joshi and Singh [67] observed a *substantial* reduction in the efficiency of 17b-HSD in mice handled with the OX1R antagonist in vivo, but not in vivo. In addition, it appears that the antagonist has no consequences on the efficiency of 3b-HSD under in vivo conditions, while enzymatic activity was significantly improved under ex vivo conditions. These dissimilarities in enzyme activity between in vivo and ex vivo are partly attributable to discrepancies in the biological compounds under the two dosing conditions [74]. Furthermore, it has also been shown that it is methodologically complicated to fully imitate all the effects of cytosol under in vivo conditions when considering ex vivo conditions [74]. In addition, the authors [67] observed an increase in serum and intratesticular levels of E2, together with an increased expression of P450arom, despite the lowest levels of T being detected in OX1R antagonist-handled mice. It is hard to elucidate the increased estrogen level despite the decreased testosterone level as the latter acts as a substrate for E2. The rise in the expression of P450arom in the male gonad after antagonist healing may have contributed to the increase in the level of E2. As reported in the literature, dehydroepiandrosterone (DHEA) is transformed into 3b-HSD androstenedione and/or 17b-HSD androstenediol in steroidogenic machinery [75]. As is currently known, therefore, it is conceivable that DHEA should be quickly converted into androstenedione rather than androstenediol due to an increase in 3b-HSD expression. In detail, androstenedione is transformed into testosterone by 17b-HSD and then into testosterone and estradiol by the enzyme P450arom [75]. In this regard, there was an increase in the expression of P450arom and a consequent reduction in 17b-HSD after treatment with the OX1R antagonist, resulting in a growth in the level of E2 and a drop in the level of T. However, the exact mechanism requires further study. It is common knowledge that the maintenance of spermatogenesis is firmly related to the intricate balance of the hypothalamic–pituitary–testicular (HPT) axis, and the dispensation or removal of estrogen can influence the HPT axis [70]. Thus, any interference (i.e., deprivation or excess) in the level of E2 could cause an alteration in the HPT axis, generating adverse effects on spermatogenesis [70,76]. Lipids present in the juxtamembrane represent the main targets of reactive oxygen reagents, which are the basis of lipid peroxidation (LPO) [77]. SOD and catalase are essential parts of the cell’s antioxidant defense system against oxidative stress. Oxidative stress causes alterations in steroidogenic action, thus causing damage to testicular functions [78]. Therefore, it is demonstrable that changes in the hormonal environment in the testis cause oxidative stress to rise and the apoptosis of germ cells, occurring in the inhibition of spermatogenesis [79]. In the study by Chaki et al. [79] in rats, the amplification of oxidative stress caused by an increment in LPO and a reduction in SOD and catalase activity under antagonist treatment may have originated from (a) changing levels of T and/or E2, (b) the impact of antagonist healing, or (c) by concerted effects (a and b). More pronounced oxidative stress in the male gonad after antagonist treatment might result in the impairment of spermatogenesis. It was confirmed that, in P mice, the blockage of OXA/OX1R binding in the testis was due to the inhibition of steroidogenesis, with a decrease in the T level and an increase in the E2 level. In addition, treatment also causes an increase in the LPO level and a decrease in the activities of antioxidant enzymes (i.e., SOD and catalase) in the male gonad. Consequently, an alteration of hormonal levels and a rise in oxidative stress in the testis determine spermatogenesis defects. In conclusion, it may be argued that OXA/OX1R in the male gonad intervenes in the modulation of murine steroidogenesis and spermatogenesis (normal localizations are reported in Table 1).

## 4. The Orexinergic System in the Male Genital Tract of Ruminants

### 4.1. Sheep Male Genital Tract

The schematic representation of the distribution of the orexinergic complex is summarized in Table 1. The OX1R mRNA in sheep has sequence homology of 91% and 87% with that of humans and rats, respectively, sharing 92.1% homology in their amino acid sequence [4]. However, only OX1R-mRNA and not OX2R-mRNA has been described in sheep testes [4].

Sheep differ from humans and rodents in their digestive and metabolic systems [80,81,82,83,84]. Sheep have seasonal variability in various organic functions such as appetite, food intake, and body weight, and they have a different digestive physiology to monogastric species. Consequently, the distribution of orexin receptors is assumed to differ significantly between sheep, humans, and rodents. In addition, the amount of food ingested deeply affects the physiology of the reproductive system in adult male sheep [83]. Thus, more extensive studies are necessary.

*Testis:* In mature rams and bucks (but not bulls), food intake has an effect on the size of the testicle, especially on the number of seminiferous tubules, thus influencing the spermatogenic capacity of the organ. Therefore, spermatogenic efficiency may be impaired, particularly regarding the rate of sperm production. Excluding desperate malnutrition, these effects are not related to changes in testosterone production or sexual behavior. Although central pathways to these processes have not yet been elucidated, it has been suggested that orexinergic systems are involved [83,84,85]. Discrepancies have been found in the expression levels of OX1R and of OX2R, the latter being very low in the sheep pineal gland, as opposed to the rat, where only OX2R mRNA was detected [86]. In these two species, there may be anatomo-physiological dissimilarities in the regulation of reproductive function by photoperiod on the pineal gland [84]. As occurs in rats and humans, both OX1R and OX2R have been described in the ovine hippocampus, amygdala, and olfactory bulb, indicating an orchestrated role of the orexinergic complex in cognition and emotion [87,88,89]. OX1R and OX2R genes have been detected in the sheep pituitary gland, as previously described in humans and rats. The expression of OX1R—but not that of OX2R—has also been observed in the ovine adrenal gland, in contrast to that described in rats and in humans, where both receptors have been shown to be expressed [21,90]. The lack of OX2R mRNA in the sheep adrenal gland could be ascribed either to species differences or, alternatively, to methodological dissimilarities. The detection of OX1R mRNA in the adrenal gland and testis indicates that circulating OXs in sheep may have a direct action on the adrenal and gonadic axis through OX1R in target organs. However, the physiological impacts of circulating OXs on peripheral organs should be further explored (normal localizations are reported in Table 1).

### 4.2. Cattle Male Genital Tract

In these species, only the urethral–prostatic tract has been studied.

*Prostate*: PPO, OXA, and OX1R have been found. OXA and OX1R immunoreactivity (ir) was localized in the exocrine cells, but not in the endocrine cells (chromogranin-positive), with a different intensity of staining in all of the subjects studied [43].

*Urethra*: OXA peptide immunoreactivity was found in a sub-population of endocrine cells (chromogranin-positive) in a small number of subjects [43] (normal localizations are reported in Table 1).

### 4.3. South American Camelid Alpaca Male Genital Tract

The South American camelid alpaca belongs to the genus Vicugna, along with the vicuna (*Vicugna vicugna*), and has phylogenetical similarity to llama (*Lama glama*) and guanaco (*Lama guanicoe*) [91]. These animals are widespread in the Andean states of South America, from Ecuador to southern Chile. Little is still known about the reproductive characteristics of the alpaca. However, the transition from colder Andean areas to areas with a milder climate has increased the fertility of these species, which have thus become increasingly richer non-seasonal breeders [92].

The alpaca testis expresses PPO [91]. In particular, OXA has been found in numerous Leydig cells and rarely in Sertoli cells, spermatogonia, pre-leptotene spermatocytes, and spermatids. OX1R has been found in Leydig cells and in mature spermatids, with a distribution similar to that of the OXA peptide [91].

OXB has been described in spermatogonia, spermatocytes, and mature spermatids. It does not coexist with OXA. OX2R was described in Leydig cells and spermatids [91,93,94]. In in vitro experiments, OXA improved T synthesis, while the supplement of MIS reduced it crucially. The steroidogenic effect of OXA was more pronounced with longer exposure, as well as after the addition of MIS. In contrast, longer exposure to MIS after OXA addition led to a steroidolitic effect. These results represent conclusive proof that OXA—by linking OX1R and MIS—appears to play an antagonistic role in the determination of OXA-evoked testicular steroidogenesis. The presence of the OXA antagonist SB-408124 in OXA-treated slices led to a strong decrease in the steroidogenic effect of the protein. Taking these findings together, the authors [91] speculated that the mammalian seminiferous compartment contains a cellular complex which is capable of secreting and/or internalizing OXA. A pivotal role is played by the Sertoli cell, which has already been defined as the main source of OXA in rat testes [35]. Testicular slices in vitro treated with OXA resulted in a sharp reduction in the production of the mRNA of the stem cell factor (SCF) [16]. SCF is a substance secreted by the Sertoli cell which, when released at the intercellular level, causes stimulation in the proliferation of spermatogonia and, consequently, an increase in the development of germinal epithelium [95,96]. Therefore, the intratubular synthesis of OXA could indirectly decrease the proliferation of germ cells, leading to the inhibition of SCF biosynthesis [42]. Thus, a direct impact of OXA on the germ compartment *cannot be ruled out*, as different testicular cytotypes express OX1R. There is a clear similarity between orexinergic cytotypes between alpaca and rat testes. Although the two animal species differ from a phylogenetic point of view, it has been suggested that the complex of cells containing OXA and OX1R is essential for the functioning of the mammalian testis [13]. The presence of OX1R in mammalian Leydig cells indicated a possible function of OXA in testicular steroidogenesis [34,35,91]. In particular, it has been hypothesized that OXA, primarily synthesized by the Sertoli cells, acts directly on the same cells by the inhibition of autocrine function and, therefore, the production of Müllerian inhibitory substance (MIS), also secreted by Sertoli cells, which has a steroidolitic action. The stimulation of testosterone secretion has been experimentally demonstrated through the in vitro sequential implementation of OXA and MIS (and vice versa) to tissue slices from the testes of rats and alpacas [34,91].

While the stimulation of steroidogenesis OXA-induced has been corroborated, conversely, OXB appears to have no effect on testosterone secretion. Liguori et al. [13] demonstrated the clear analogy between the orexinergic systems of the alpaca and rat testicles, and it could therefore be hypothesized that this system intervenes in the regulation of spermatogenesis, as OXA and OX1R also occur in the main cells of the epididymis head [97] (normal localizations are reported in Table 1).

## 5. The Orexinergic System in the Male Genital Tract of Dog

PPO distribution has been observed in canine normal and cryptorchid testis and epididymis [94].

*Testis*: In particular, OXA and OX1R have been found in the Leydig cells of normal subjects, and in Leydig cells, Sertoli cells, and early germ cells in the cryptorchid subjects. In contrast, OXA alone occurred in the early germ cells [94], while OXB was found in a sub-population of Leydig cells (intermingled among the negative ones), in pachytene and zygotene spermatocytes, in secondary spermatocytes, and from immature to mature spermatids in the seminiferous tubule of normal dogs. In contrast, OXB occurred in Sertoli and Leydig cells and rarely in early gonocytes of cryptorchid dogs [15].

OX2R-IR has been found in Leydig cells, in pachytene spermatocytes, in diplotene and zygotene spermatocytes, and in secondary spermatocytes throughout spermatid maturation. OXB- and OX2R-IR have been described in all stages of the germ-developing cycle by Soares et al. [98]. In the cryptorchid testis, OX2R occurred mainly in Leydig and Sertoli cells [15].

The widespread expression of OXB and OX2R in numerous cytotypes in the testis of healthy dogs and their localization pattern closely resembled the previous results in rats and alpacas [13,14], allowing for the hypothesis of a potential role in the regulation of spermatogenesis and endocrine functions in the canine testicle through a mechanism involving autocrine and/or paracrine action. In the retained gonad from cryptorchid dogs, the distribution of OXB and OX2R was limited to Leydig and Sertoli cells, which may be ascribed to damaging consequences of the undescending testis on spermatogenesis, leading to infertility [99,100,101,102]. In addition, OXB/OX2R did not appear to have a direct steroidogenic effect on the testis, as reported in the normal rat male gonad too [13]. Conversely, steroidogenic activity was detected when OXA bound OX1R [94,103]. OXA/OX1R binding determined the stimulation of T secretion in rats [34], alpacas [91], mice [8,9,10,14], and canine normal and cryptorchid testes [94]. In particular, in the testes of normal and cryptorchid dogs, the stimulation of T synthesis was OXA-mediated, and the consequent reduction in the production of 17bE was induced by the inhibition of ARO activity evoked by OXA.

Following these observations, the action of OXB on Leydig cells is a subject still much debated, although OXB may be implicated in other Leydig cell activities or act on paracrine-induced spermatogenesis [13,14].

In addition, OXB and OX2R have been evidenced in seminiferous and interstitial compartments of dog testes [15].

*Epididymis*: OXB has been found in all segments of the principal cells, both in normal and cryptorchid animals. In contrast, OX2R was only localized in the efferent ductules in the head of normal and cryptorchid epididymis [15].

Although OX2R has been observed in the corpus and cauda of both the normal and cryptorchid epididymes, the receptor was undetectable immunohistochemically. This discrepancy could be attributed to (a) enhanced sensitivity of molecular procedures than immunohistochemistry or (b) high turnover rate of the production/internalization of OXs. Taking into account the null effect of OXB-binding OX2R on steroidogenesis, other roles have been assumed by these substances in the testis and epididymis. The location that is supranuclear of OXA and OXB in the major epididymal cells established that these compounds could be detected at different sites, thus acting in a paracrine-wise manner. As has been reported by Crabo et al. [104], the proximal regions of the epididymis and the efferent ductules are involved in the consumption of 90% of the liquids released by the seminiferous tubules. The epididymis head and efferent ductules have a cytoplasmic location of OXB and OX2R, which would allow for the hypothesis that the fluid generated in the testis could then undergo a re-absorption process in the head of the epididymis and in the efferent ductules. This suggests that OX2R/OXB binding may regulate the processes of absorption and secretion in the proximal portions of the epididymis.

## 6. The Orexinergic System in the Male Genital Tract of Boar

In these species, only the testis was studied [105]. The authors reported, immunohistochemically, the presence of OX1R in round, oval, and elongated spermatids in the seminiferous tubules of boar testes. All immunostained spermatids were at the same stage of the developmental cycle. In contrast no immunoreactivity, interstitial compartments were observed.

**Table 1 vetsci-11-00131-t001:** The orexinergic complex occurring in the mammalian genital tract.

Location	Human	Rat	Mouse	Ram	Bull	Alpaca	Dog	Boar
Testis Leydig cells Sertoli cells TPMCs Germinal epithelium Interstitium tubular compartments	OX1R, OX2R [17]	PPO [16], OXB, OX2R [13], OXA [12,16], OX1R [34]	OXA, OX1R [8], OXB, OX2R [7]	OX1R [4]		PPO [91], OXA [91], OX1R [91], OX2R [14], OXB [14]	PPO, OXA, OX1R [94], OXB [15], OX2R [15]	OX1R [105]
Epididymis	PPO, OX1R, OX2R [17]	PPO, OXA, OX1R [11], OXB, OX2R [46]				OXA, OX1R [97]	OXB, OX2R [14]	
Seminal vesicles	OX1R, OX2R [17]							
Urethra					OXA [43]			
Prostate	PPO, OXA, OX1R [18], OX2R [31]				PPO, OXA, OX1R [43]			
Penis	PPO, OX1R, OX2R [17]							

**Table 2 vetsci-11-00131-t002:** The orexinergic complex occurring in the most common pathologies of the mammalian genital tract.

Location	Human	Dog
Testis	-	Cryptorchidism: OXA, OX1R [94], OXB, OX2R [15]
Prostate	Iperplastic prostate: OXA, OX1R [19], OX2R [31] Benign iperplasia: OXA, OX1R, OX2R [31] Cancer prostate OX1R, OX2R [18]	-
Epididymis	-	Cryptorchidism: OXB, OX2R [15]

## 7. Insights and Future Directions

The treatment of data related to the role of the orexinergic complex in the pathogenesis of acute/chronic inflammation is notable. In more detail, OXs and their receptors were detected in several forms of inflammations of the gastro-intestinal tract, such as intestinal bowel disease (IBD), including ulcerative colitis (UC) and Crohn’s disease (CD), pancreatitis, liver fibrosis, etc., as well as in multiple sclerosis (MS) and septic shock [106]. As reported in the literature, chronic inflammations represent one of the high risk factors of digestive cancer [107]. OX1R was found in inflamed mucosa from patients affected by UC and CD [108] and colon cancer, but not in normal human intestinal epithelium [109]. Thus, OXA displays anti-inflammatory properties in IBD and UC demonstrated by injecting OXA intraperitoneally to the DSS-induced colitis mouse model which ectopically expresses OX1R [108]. The anti-inflammatory responses of OXA are expressed through the inhibition of various cytokines, including IL-6, TNFα, IL-8, IL-1β, IL-1α, IL-17, and MCP-1 cytokines mediated by OX1R via PLC signaling pathways [108]. Similarly, OxA was also able to reduce, in MS, the activation of NF-κB signaling pathways in the tissue inflammation site [110].

Considering the anti-inflammatory impact of OXA/OX1R as just described, there are currently no data in the literature on the presence of the orexinergic complex in organs belonging to the male genital system, suffering from inflammatory diseases (i.e., prostatitis, orchitis, epididymitis). The only data concern the evidence of mRNA and OX2R peptide, but not of PPO and OX1R in the normal human prostate and in that affected by BPH [31]. In particular, BPH is nothing more than a common pathology in adult men, likely the consequence of inflammatory phenomena [111].

Moreover, one cannot fail to mention the cryptorchidism condition, which is one of the causes of infertility and testicular germ cell cancer in men [112]. Despite the unclear molecular mechanisms, it seems that the retention of the abdominal gonad’s high testicular temperature leads to the depletion of germ cells and apoptotic phenomena. Haploid germ cells of the next stage appear to be the most susceptible to increased heat [112]. Leydig and Sertoli cells appear to be quite resistant to temperature increases in the gonad and, contrary to popular belief, are subject to hyperproliferation. Gonad retention has spermatogonial disorders, which is the most common cause of atypical germ cell differentiation and neoplastic transformation of testicular germ cells. However, under physiological conditions, OXs can act to improve cell proliferation and survival [113]. The binding of OXs with their cognate receptors is responsible for extensive apoptosis and a decrease in cell growth in numerous neoplastic cell lines, such as human colon cancer cells and human neuroblastoma cells [114], rat pancreatic cancer cells [115], rat glioma cells C6 [116], and Chinese hamster ovary transferred cells (CHO) with cDNA OX1R. Orexin is responsible for inducing this apoptotic mechanism that may be associated with the release of cytochrome C from the mitochondria and activation of caspase-3/7 in an OX1R-evoked manner [117]. In addition, OXA can trigger apoptosis by binding OX2R in rat pancreatic cancer cells and rat C6 glioma cells [116,117]. From this assumption, it can be assumed that the elevated levels of PPO and orexin receptor mRNAs in retained organs may be induced by OXs. In particular, the expression of OX1R and/or OX2R seems to play a role in modulating pre-cancerous evolution, and the activation of apoptosis triggered by orexin receptors.

Autophagy is another event that occurs during cryptorchidism which is associated with the thermal rise that occurs in the retained gonad and consequently damages testicular spermatogenesis [118,119]. In the literature, it has been described that OXA brought about autophagy through the ERK machinery in human colon cancer cells HCT-116. Apoptosis and autophagy are simultaneous events involved in stimulating the death of testicular germ cells. Cryptorchidism also causes epididymal abnormalities [120,121], with abnormalities in the mitochondrial organelles. Spermatic mitochondria also cause the production of reactive oxygen species (ROS), which is the basis of oxidative injury [122,123]. ROS enhanced autophagy in vitro [124] and apoptosis in spermatozoa [122,123]. A clear relationship between OXB and its redox control has also been described in the ovarian follicles of pigs [124]. From what has been described in the literature, it can be hypothesized that the high amount of PPO and OX2R genes detected in the testis and head epididymis, compared to the OXB/OX2R ligand, may modulate the redox state by improving autophagy induction in the testis and pro-apoptotic events in the head of the epididymis. Moreover, an increase in heat shock protein-70 (HSP 70) transcript expression has been described in a cryptorchid corpus, cauda epididymis, and in the vas deferens from human beings [125].

From these latest studies, conducted in dogs under normal conditions and cryptorchidism, it can be concluded that the orexinergic system is involved in the regulation of testicular spermatogenesis, as well as epididymal absorption and secretion activity. In fact, OXB binding OX2R may determine the induction of autophagy at the testicular level and/or induce a pro-apoptotic phenomenon in the head of the epididymis by modulating the redox state, while OXA binding OX1R may modulate the expression of HSP 70 at the level of the body epididymis [15]. The studies highlighted in this narrative review should be taken into consideration when developing other experimental procedures regarding the potential regulation of the male reproductive functions OX-mediated in health and disease states (normal and pathological localizations are reported in Table 1 and Table 2).

The cause of male infertility can be attributed to non-genetic factors including environmental pollution (organic, inorganic, and air pollutants), occupational exposure (high temperature, organic solvents, and pesticides), and poor lifestyle (diet, sleep, smoking, alcohol consumption, and exercise), as determined by epigenetic studies. Environmental pollution seems to have particular effects on the male reproductive system, specifically spermatogenesis. Human semen is an early and sensitive indicator of environmental pollution [126]. In addition to psychological distress and economic constraints, poorer semen quality could lead to heritable alterations, such as congenital structural abnormalities that, in addition, predispose to late-onset diseases in adulthood. Genetic and epigenetic mutations might cause irreversible damage to spermatogenesis. Similar factors might be responsible for the raising of prostatic cancers in men too. It is well known that that male infertility could be attributable, therefore, to sperm DNA damage and epigenetic alterations affecting spermatogenesis. Further studies are needed to better understand the effects of intrinsic or extrinsic factors on the pathogenic and molecular processes that determine infertility or prostate cancer.

Moreover, as described above, the orexinergic system would act on prostate cancer, inducing pro- or anti-apoptotic responses [18,19,20,32]. OXs might represent a promising anti-neoplastic and/or biomarkers for tumor risk/prognosis. In order to better understand the therapeutic and/or diagnostic/prognostic validity, other studies are necessary that include, however, new and more suitable animal experimental models. In this regard, this was suggested by Costagliola et al. [32], who postulated clinical similarities existing between canine and human prostate cancer. In fact, the genes and pathways involved in carcinogenesis are homologous between dogs and humans. Just like human prostate cancer, canine prostate cancer also tends to develop bone metastasis, making it a valuable experimental model for human prostate cancer research. Unlike humans, dogs have a common prostate cancer that is androgen-independent, which makes them an excellent experimental model for human androgen-independent prostate cancer.

The environmental impact on the development of such diseases is increasingly recognized, as well as new preventive strategies to counteract and/or modulate the effects of pollution on human health [127,128,129]. New studies should, however, be carried out in order to better analyze the problems described above.

## 8. Conclusions

In recent decades, the study of OXs in pathophysiological settings has assumed great importance, especially regarding reproduction, although the role of OXs is currently understood only to a limited extent. Data obtained in several mammalian species, including humans, clearly indicate that the orexin system occurs in the cellular populations throughout the male genital tract, particularly in the testis. The similarity of the testicular cytotypes localizations and relative functions further confirms the hypothesis of the roles of this system on spermatogenesis and steroidogenesis. Further studies concerning both the expression and functions of the orexin system are necessary, including those considering the species-specific hormonal regulation of mammalian reproductive processes. OXs regulate the HPG axis and act directly on the gonads via autocrine/paracrine signaling.

Taking into account the involvement of OXs in the regulation of food intake and energy homeostasis, a link between energy reserves and the reproductive capabilities of the organism cannot also be ruled out.

In conclusion, the importance of the peripheral impact of the orexinergic system led us to hypothesize its potential interest as therapeutic targets and/or biomarkers for risk/prognosis in a wide range of human and animal pathologies, including acute/chronic inflammation, infertility, and cancers.

## Data Availability

Not applicable.

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
