# Peer review of "A Comparative Analysis of Orexins in the Physio-Pathological Processes of the Male Genital Tract: New Challenges? A Review"

_vetsci, 2024, doi:10.3390/vetsci11030131_

Round 1

Reviewer 1 Report

Comments and Suggestions for Authors

This is an interesting Review paper. The topic is within the scope of the journal. The data on the role of orexins in the physio-pathological processes of the male genital tract are of high interest.  This review highlights the studies on the role of orexins the male genital tract in humans and animals and discusses their potential use as markers for prostate cancer and a therapeutic target for male infertility. Orexins A and B (hypocretins) are two neuropeptides produced in the hypothalamus acting by their specific receptors, receptor 1 and 2. These neuropeptides are responsible for control of food intake, energy expenditure, sleep-wake function as well as the regulation of pituitary hormone secretion and reproduction. Growing evidences indicates that orexins are involved in steroidogenesis and spermatogenesis in testes and maturation of spermatozoa in the epididymis. Orexins also appear to play a role in prostate cancer. The manuscript is well written and can be published in its present form. I only suggest a more extensive discussion on the involvement of orexins in prostate cancer in humans and probably dogs.

Author Response

Comments of the Reviewer #1:

This is an interesting Review paper. The topic is within the scope of the journal. The data on the role of orexins in the physio-pathological processes of the male genital tract are of high interest. This review highlights the studies on the role of orexins the male genital tract in humans and animals and discusses their potential use as markers for prostate cancer and a therapeutic target for male infertility. Orexins A and B (hypocretins) are two neuropeptides produced in the hypothalamus acting by their specific receptors, receptor 1 and 2. These neuropeptides are responsible for control of food intake, energy expenditure, sleep-wake function as well as the regulation of pituitary hormone secretion and reproduction. Growing evidences indicates that orexins are involved in steroidogenesis and spermatogenesis in testes and maturation of spermatozoa in the epididymis. Orexins also appear to play a role in prostate cancer. The manuscript is well written and can be published in its present form. I only suggest a more extensive discussion on the involvement of orexins in prostate cancer in humans and probably dogs.

Answer: We thank the Reviewer for his/her very careful reading of our manuscript and for his/her important and kind comments. As suggested by the reviewer, we have provided to add sentences in the Discussion section focusing the involvement of orexins in prostate cancer in humans and probably dogs (lines 659-670).

Reviewer 2 Report

Comments and Suggestions for Authors

The authors examine the experimental evidence collected on different species (human, rat, mouse, ram, bull, dog and alpaca, but curiously not on boar) from the end of the 90s to today to support the male reproductive aspects of orexins. They cite 123 works of which only 12 are from the last 5 years, while 22 are self-citations.

They give particular emphasis to the description of orexins expression patterns and potential functional roles in physiological and pathological conditions, which suggest that orexins can operate as neuroendocrine regulators on spermatogonesis and steroidogenesis.

A similar conclusion, i.e. that expression of orexins in the testis is cell-, developmental-and stage-specific, and they operate as autocrine/paracrine regulators of testicular function” was already made by Nurmio et al. 2010 in a review titled “Orexins and the regulation of the hypothalamic-pituitary-testicular axis”, that however, is not cited in the present manuscript. Nurmio et al., (2010) added the hypothesis of a potential potential role of orexins as a link between energy balance and reproduction. This possible link is not considered in the present review.

 The authors give a rich list of the functions associated with the orexinergic system and hypothesize the potential of such peptides as a therapeutic goal for a number of reproductive disorders and therefore the processing of drugs for the treatment of male hypo/infertility and cancer.

The gaps in knowledge are identified, but often in a very generic way and in relation to each individual publication.

The work could benefit from clearer organization and possibly some formatting improvements to enhance readability. Infact, although the authors themselves conclude that “the similar occurrence in the testicular cytotypes and physiological studies in different species presenting similar functions allow for the hypothesis of a similar role of this system on spermatogenesis and steroidogenesis”, the results of the various works cited are often treated only sequentially one after the other, without an obvious common thread between the aspects investigated or the results obtained, and sometimes without even starting a new paragraph when the topic changes. The analysed studies are grouped by species, and within the species, by organ, but sometimes different works that have dealt with orexin in the same type of cells and in the same species are not even compared but simply listed, interspersed with other articles. All this, added to the lack of figures, images or diagrams, makes reading a little tiring and unappealing.

The tables summarizing the topographic distribution of the orexinergic complex in healthy mammalian male genital tract and in its most common pathologies, might be completed adding a column with the correspondent references.

Many acronyms are not preceded by the full name, while the acronyms for orexin should be standardized (sometimes OX, sometimes OR).

The quotations in the text start from n. 2

There are several typos throughout the text that require spell checking before printing.

Comments on the Quality of English Language

The quality of the English language is good.

I only have 3 small suggestions:

Line 210 tested should be replaced with “to test”

Line 312 “a pivotal roles”: please correct with “a pivotal role”

Line 563 to make the exposition of the points in the list more homogenous, “b) the production/internalization of orexins has a high turnover rate” could be replaced with “high turnover rate of the production/internalization of orexins”

Author Response

Comments of the Reviewer #2

GENERAL

The authors examine the experimental evidence collected on different species (human, rat, mouse, ram, bull, dog and alpaca, but curiously not on boar) from the end of the 90s to today to support the male reproductive aspects of orexins. They cite 123 works of which only 12 are from the last 5 years, while 22 are self-citations.

Answer: As kindly suggested, we have provided to add more recent bibliographic entries and we have provided to create another paragraph entitled: “The orexinergic system in the male genital tract of pig”

They give particular emphasis to the description of orexins expression patterns and potential functional roles in physiological and pathological conditions, which suggest that orexins can operate as neuroendocrine regulators on spermatogonesis and steroidogenesis.

A similar conclusion, i.e. that expression of orexins in the testis is cell-, developmental-and stage-specific, and they operate as autocrine/paracrine regulators of testicular function” was already made by Nurmio et al. 2010 in a review titled “Orexins and the regulation of the hypothalamic-pituitary-testicular axis”, that however, is not cited in the present manuscript. Nurmio et al., (2010) added the hypothesis of a potential potential role of orexins as a link between energy balance and reproduction. This possible link is not considered in the present review.

Answer: Thank you the referee for the kind suggestions, we have provided to add Nurmio et al., (2010) as new bibliographic entries by adding the hypothesis of a potential potential role of orexins as a link between energy balance and reproduction (lines 64-69; 172-176; 191-193; 697-698).

The authors give a rich list of the functions associated with the orexinergic system and hypothesize the potential of such peptides as a therapeutic goal for a number of reproductive disorders and therefore the processing of drugs for the treatment of male hypo/infertility and cancer.

The gaps in knowledge are identified, but often in a very generic way and in relation to each individual publication.

The work could benefit from clearer organization and possibly some formatting improvements to enhance readability. Infact, although the authors themselves conclude that “the similar occurrence in the testicular cytotypes and physiological studies in different species presenting similar functions allow for the hypothesis of a similar role of this system on spermatogenesis and steroidogenesis”, the results of the various works cited are often treated only sequentially one after the other, without an obvious common thread between the aspects investigated or the results obtained, and sometimes without even starting a new paragraph when the topic changes. The analysed studies are grouped by species, and within the species, by organ, but sometimes different works that have dealt with orexin in the same type of cells and in the same species are not even compared but simply listed, interspersed with other articles. All this, added to the lack of figures, images or diagrams, makes reading a little tiring and unappealing.

Answer: We thank you the referee for his/her kind suggestions for improving the manuscript. We have clarified and re-organized the periods in order to improve the ease of reading. Additionally, the data content of Tables 1 and 2, were also improved and re-organized by adding the relative references, as requested.

The tables summarizing the topographic distribution of the orexinergic complex in healthy mammalian male genital tract and in its most common pathologies, might be completed adding a column with the correspondent references.

Answer: As kindly suggested, we have provided to add a column with the correspondent references to the Tables.

Many acronyms are not preceded by the full name, while the acronyms for orexin should be standardized (sometimes OX, sometimes OR).

Answer: As kindly suggested, we have provided to standardized the acronyms for orexin to OX.

The quotations in the text start from n. 2

Answer: We apologize first for the carelessness and thank the Referee for the suggestion. We have provided to start from n.1

There are several typos throughout the text that require spell checking before printing.

Answer: As suggested by the referee we have provided to carefully review the manuscript and checking the typos.

Comments on the Quality of English Language of the Reviewer #2

The quality of the English language is good.

I only have 3 small suggestions:

Line 210 tested should be replaced with “to test”

Line 312 “a pivotal roles”: please correct with “a pivotal role”

Line 563 to make the exposition of the points in the list more homogenous, “b) the production/internalization of orexins has a high turnover rate” could be replaced with “high turnover rate of the production/internalization of orexins”

Answer: We thank the reviewer # 2 for his/her appreciation of the quality of the manuscript. As kindly suggested, we have provided to correct the minor revisions.

Reviewer 3 Report

Comments and Suggestions for Authors

Appetite, obesity and female reproduction are more discussed topics, but the relationship between male obesity and reproduction is rarely discussed. This article is fascinating. In ancient China, there is a saying that "full food thinks lust", which means that there is sexual desire when you are full, so it can be seen that there is a natural connection between appetite and sexual desire!

It would be even better if some of the sections Orexins and orexin receptors associated with inflammation were added.

Author Response

Comments of the Reviewer #3:

Appetite, obesity and female reproduction are more discussed topics, but the relationship between male obesity and reproduction is rarely discussed. This article is fascinating. In ancient China, there is a saying that "full food thinks lust", which means that there is sexual desire when you are full, so it can be seen that there is a natural connection between appetite and sexual desire!

Answer: We thank the reviewer # 3 for his/her appreciation of the quality of the manuscript. As suggested, for greater clarity and ease of reading, the paragraph entitled "Future Directions" was renamed "Insights and Future Directions" where some sentences regarding the natural connection between appetite and sexual desire were cited (lines 699-701).

It would be even better if some of the sections Orexins and orexin receptors associated with inflammation were added.

Answer: We thank the reviewer # 3 for his/her appreciation of the quality of the manuscript. As suggested, for greater clarity and ease of reading, the paragraph entitled "Future Directions" was renamed "Insights and Future Directions" where some sentences regarding the involvement of orexins in inflammation were added (lines 572-593; 702-705).
